# BmCDK5 Affects Cell Proliferation and Cytoskeleton Morphology by Interacting with BmCNN in *Bombyx mori*

**DOI:** 10.3390/insects13070609

**Published:** 2022-07-06

**Authors:** Yi Wei, Xiaolin Zhou, Peng Chen, Xia Jiang, Ziyi Jiang, Zhanqi Dong, Minhui Pan, Cheng Lu

**Affiliations:** 1State Key Laboratory of Silkworm Genome Biology, Southwest University, Chongqing 400716, China; weiyi616@yeah.net (Y.W.); zhouxl@ucas.ac.cn (X.Z.); pjchen@swu.edu.cn (P.C.); jiangxia1203@yeah.net (X.J.); j942350863@gmail.com (Z.J.); zqdong@swu.cn (Z.D.); 2Key Laboratory for Sericulture Functional Genomics and Biotechnology of Agricultural Ministry, Southwest University, Chongqing 400716, China

**Keywords:** Cyclin-dependent kinase 5, *Bombyx mori*, centrosomin, cell cycle, cytoskeleton

## Abstract

**Simple Summary:**

Cyclin-dependent kinases (CDKs), as biological macromolecules, play key regulatory roles in the cell cycle process. Cyclin-dependent kinase-5 (CDK5) is part of the CDK family; however, its effect on cell cycle progression is controversial and the mechanism remains unclear. In this article, we identified a homologous gene of the Cyclin-dependent kinase family, *BmCDK5,* in the silkworm, *Bombyx mori*. We proved that the *BmCDK5* gene can regulate the cell cycle and promote cell proliferation in BmNS cells. Furthermore, we demonstrated that BmCDK5 can interact with BmCNN, and that they both affect cytoskeleton morphology but do not induce changes in microtubule protein expression; they can also promote cell proliferation.

**Abstract:**

The ordered cell cycle is important to the proliferation and differentiation of living organisms. Cyclin-dependent kinases (CDKs) perform regulatory functions in different phases of the cell cycle process to ensure order. We identified a homologous gene of the Cyclin-dependent kinase family, *BmCDK5*, in *Bombyx mori*. BmCDK5 contains the STKc_CDK5 domain. The *BmCDK5* gene was highly expressed in S phase. Overexpression of the *BmCDK5* gene accelerates the process of the cell cycle’s mitotic period (M) and promotes cell proliferation; knocking out the *BmCDK5* gene inhibited cell proliferation. Furthermore, we identified a protein, BmCNN, which can interact with BmCDK5 and represents the same express patterns as the *BmCDK5* gene in the cell cycle phase and the spatial-temporal expression of *B. mori*. This study revealed that BmCDK5 and BmCNN play roles in promoting cell proliferation and regulating cytoskeleton morphology, but do not induce expression changes in microtubule protein. Therefore, our findings provide a new insight; the *BmCDK5* gene has a regulatory effect on the cell cycle and proliferation of *B. mori*, which is presumably due to the interaction between BmCDK5 and BmCNN regulating changes in the cytoskeleton.

## 1. Introduction

Cyclin-dependent kinases (CDKs), known as serine/threonine kinases, play key roles in cell cycle regulation [1]. In general, CDK contains a regulatory cyclin subunit and a catalytic serine/threonine kinase subunit, so that it can be modulated by association with cyclins and CDKs inhibitors [2,3]. In mammals, 21 genes encode CDK according to sequence homology. The CDK family is divided into three cell cycle related subfamilies (CDK1, CDK4, and CDK5) and five transcription subfamilies (CDK7, CDK8, CDK9, CDK11, and CDK20) [1]. Cyclin-dependent kinase-5 (CDK5) is part of the CDK family and has high homology with other CDKs; however, it has unique activation modes and cellular functions compared to them [4]. Although CDK5 regulates a series of important cellular processes in cells [5,6], its effect on cell cycle progression is controversial and the mechanism remains unclear [7,8,9,10].

CDK5 was first isolated from bovine brain tissue extracts through biochemical methods [11]. CDK5 substrates include microtubule-associated protein 1B, which contains a Ksp-like motif, nerve filament protein containing KspxK τ Au and Kspxx, actin binding protein, and vesicle protein [12,13]. Studies show that CDK5 plays an important role in neuron migration, axon guidance, and synaptic formation by regulating the migration of the developing central nervous system and the cytoskeletal structure and organization of nerve cells [13,14]. Furthermore, CDK5 is related to the pathogenesis of neurodegenerative diseases such as Alzheimer’s disease (AD), amyotrophic lateral sclerosis (ALS), and Niemann’s Pick type-C disease (NPD); its hyperactivation results in the deep remodeling of the nerve cytoskeleton, and the loss of synapses eventually leads to neurodegenerative changes [15,16,17]. The current study demonstrated that CDK5 is also involved in inflammation-mediated neurodegeneration by mediating microtubule disassembly [18]. In addition, in *Drosophila melanogaster*, CDK5 regulates the timing and rate of mushroom body remodeling by dissolving the neuronal tubulin cytoskeleton; stress-induced CDK5 activity enhances cytoprotective basal autophagy [19,20]. However, the underlying molecular mechanism of CDK5 in cytoskeleton regulation function is unclear.

Microtubule, as a highly dynamic polymer, is a major component of the cytoskeleton, which is important for fundamental cellular events and developmental processes [21,22]. In addition, the centrosome is the major microtubule organizing center, which ensures orderly cell mitosis in most cell types [23]. The centrosome consists of a pair of centrioles surrounded by the pericentriolar material or matrix (PCM) [24]. Furthermore, centrosomin (CNN) is a key component of the centrosome and is abundantly distributed within the PCM [25,26]. In *D. melanogaster*, CNN plays an important role in the formation of functional centrosomes and is also essential for the organization of actin and microtubule cytoskeleton [27,28].

In this article, we identified a silkworm homologous gene of the Cyclin-dependent kinase family, *BmCDK5*. We proved that the *BmCDK5* gene can regulate the cell cycle and promote cell proliferation. Additionally, we identified that BmCNN can interact with BmCDK5. Furthermore, we demonstrated that BmCDK5 and BmCNN both affect cytoskeleton morphology, but do not induce expression changes in microtubule protein; they can also promote cell proliferation.

## 2. Materials and Methods

### 2.1. Gene Identification and Homology Analysis

All gene sequences were obtained from the National Center for Biotechnology Information (NCBI). Sequence alignments were constructed using GeneDoc 3.0 software. Protein domains were predicted by SMART software (http://smart.embl-heidelberg.de/; accessed on 19 May 2020). The phylogenetic tree was built using MEGA 6.0 software. Primer Premier 5.0 software was used to design the primers. The sgRNAs for the knock out cloning vector were designed by CRISPR direct (http://crispr.dbcls.jp/; accessed on 11 September 2019).

### 2.2. Gene Cloning

The PCR products of the target sequences and the expression vectors (Invitrogen, Waltham, MA, USA) were ligated with Solution I ligation Enzyme (Takara, Tokyo, Japan) after digestived with the same type of restriction enzymes (*EcoR I* and *Xho I* for PIZ-BmCNN-OE plasmid; *BamHI* and *EcoR I* for PIZ-BmCDK5-OE plasmid; Takara, Tokyo, Japan). The expression efficiencies or knock out efficiencies of the cloning vectors are shown in Appendix A. All primers used in this study are listed in Appendix A.

### 2.3. Silkworm Strain

Larval cDNA was extracted from third-day fifth-instar *Dazao* larvae; the silkworm strain *Dazao* was maintained at the Gene Resource Library of Domesticated Silkworm of Southwest University, Chongqing, China.

### 2.4. Cell Cultures and Cell Transfections

The cell line used in this study is BmN-SWU1 (BmNS) from the ovary of *Dazao* strain *B. mori*, which were cultured in insect medium (TC-100, United States Biological, MA, USA) at 27 °C. In the transient transfections experiment, we used the TransIT^®^-LT1 transfection reagent (Mirus, Beijing, China) to transfect plasmids into cells using a ratio of 1 μL:2 μg when the cell density was 80%.

### 2.5. Flow Cytometry

Cells were incubated with cell cycle detection reagent (Beyotime, Shanghai, China) at 37 °C for 30 min after fixing in 70% ethanol at 4 °C overnight after 48 h of transfection. Cell cycle phase changes were analyzed immediately using flow cytometry (CYTOFLEX, Beckman Coulter, Brea, CA, USA).

### 2.6. 5-Ethynyl-2′-deoxyuridine (EdU) Assay

EdU is a thymidine kinase analogue that inserts the thymidine kinase (T) into duplicated DNA during cell proliferation; it can be visually detected after reacting with Apollo fluorescent dyes for EdU. In this study, cell proliferation activity was detected via EdU lablling (Beyotime, Shanghai, China) after 48 h of transfection; green fluorescence represented EdU positive cells or EdU^+^ cells [29].

### 2.7. Cell Counting Kit-8 Assay (CCK-8)

The transfected cells were counted at 0 h, 48 h, and 72 h after transfection to determine cell proliferation ability using CCK-8 assay. Cells were divided into 96-well plates with 100 μL per well, with three replicates. Next, the CCK-8 solution (5 mg/mL; Biogroud, Chongqing, China) was added to each well. The well plate was placed in a 37 °C incubator for 3 h. Finally, the OD value of each sample was measured under a 540 nm wavelength to calculate the cellular viabilities.

### 2.8. Co-Immunoprecipitation (Co-IP) Assay and Western Blotting

The interacting plasmids were co-transferred into BmNS cells which were collected and lysed after transfection for 72 h. The 50 µL protein A + G Dynabeads (Life, Rockville, MD, USA) were incubated with anti-Flag/HA-tags antibody (Sigma, Rockville, MD, USA) or mouse control IgG (Beyotime, Shanghai, China) in 200 µL PBS (Phosphate Buffer Saline) for 30 min at 4 °C. Next, the lysed protein samples were added to the mixed systems for 1 h at 4 °C, after washing the Dynabeads two times with PBS. Finally, a 5 × sample loading buffer was added to the mixed systems, which were then boiled for 10 min to prepare the samples for the SDS-PAGE experiment. Following this, the protein bands were transferred onto a hydrophilic polyvinylidene fluoride (PVDF) membrane (Yeasen, Shanghai, China). The PVDF was incubated with primary antibodies and secondary antibodies (Beyotime, Shanghai, China), and visualized using a Western ECL Substrate (Bio-Rad, Hercules, CA, USA).

### 2.9. Immunofluorescence

Cells were incubated with 1% Triton X-100 (Beyotime, Shanghai, China) at 48 h after transfection for 15 min at room temperature (RT) (after fixing with 4% paraformaldehyde (Beyotime, Shanghai, China) for 15 min at RT) and then incubated with blocking buffer (Beyotime, Shanghai, China) for 1 h at RT. The primary antibodies, anti-Flag/anti-HA (1:200; Beyotime, Shanghai, China) were incubated with the cells for 1 h at 4 °C. After washing four times for 6 min using PBS, the cells were incubated with secondary antibodies (1:500; Life, MD, USA) and 4-diamidino-2-phenylindole, DAPI (1:500; Beyotime, Shanghai, China) to stain the cell nuclei. After washing four times for 6 min, the cells were observed using a confocal microscope (FV3000, Olympus, Tokyo, Japan) [30].

### 2.10. Total RNA Extraction and Quantitative Real-Time PCR

The sample was lysed by TRIzol reagent (Invitrogen, Waltham, MA, USA), and then RNA was extracted (using the RNA extraction Kit; Omega, Norcross, GA, USA), which underwent reverse transcription into cDNA using retrotranscriptional reagent (Takara, Kusatsu, Japan). After mixing the cDNA with HieffTM qPCR SYBR^®^ Green Master Mix (Yeasen, Shanghai, China), quantitative real-time PCR (qRT-PCR) was performed on the samples using the real-time PCR system (Analytik Jena, Jena, Germany) with sw22934 as the internal control [31].

### 2.11. Statistical Analysis

All experiments were performed at least three times with biological replications; the data were stated as the mean ± SD of three independent experiments. The student’s *t*-test was used to determine all statistically significant differences between treatments. The notation “ns” indicates “not statistically significant”, or *p* ≥ 0.05; “*” indicates “statistically significant” or 0.01 ≤ *p* < 0.05; and “**” indicates “highly significant” or *p* < 0.01 [31].

## 3. Results

### 3.1. Cloning and Identification of BmCDK5 Gene

A full-length *BmCDK5* gene was cloned from the larval cDNA of the *Dazao* strain, which encompassed a coding sequence (CDS) of 897 bp encoding a protein of 298 amino acids. Multiple sequence analysis demonstrated that *BmCDK5* is highly conserved in general (Figure 1A). Phylogenetic analysis showed that *BmCDK5* and *Tribolium castaneum CDK5* could be clustered together (Figure 1B). These results suggest that *BmCDK5* is a member of the CDK5 family.

### 3.2. Expression Pattern of BmCDK5 Gene

The expression levels of *BmCDK5* were examined in the hemolymph, epidermis, head, testis, ovary, malpighian tubule, trachea, midgut, fat body and silk gland of the third day of fifth instar larvae (Figure 2A). We found that *BmCDK5* was expressed in all larvae tissues, with the highest expression level in the gonads. It was highly expressed in the third day of fourth instar larvae, in the early and after wandering stages of the growth period (Figure 2B). Meanwhile, rigorous cell proliferation and replication occurred in the reproductive organs, which grow rapidly in the pupal stage. As a cyclin-dependent kinase, *BmCDK5* was highly expressed in silkworm testes and ovaries, which indicates that *BmCDK5* is particularly necessary for cell proliferation of the reproductive organs. Furthermore, we analyzed the expression of *BmCDK5* in different cell cycle phases. The cell cycle was blocked at G1, S, and G2/M phases after incubating the cells with synchronous reagents Aphidicolin, Hydroxyurea (HU), and Nocodazole, respectively (Figure 2C). *BmCDK5* was highly expressed in S phase (Figure 2D).

### 3.3. Overexpression of BmCDK5 Gene Promotes Cell Proliferation in BmNS Cells

The *BmCDK5* gene was overexpressed in the BmNS cells to determine its regulation in cell cycle progression. We transfected the recombinant plasmid for overexpressing *BmCDK5* (BmCDK5-OE) into cells, and assessed the expression levels of *BmCDK5* using qRT-PCR at 48 h after transfection (Appendix A). Following this, flow cytometry (FCM) analysis was performed to determine the cell cycle. Overexpression of *BmCDK5* did not significantly reduce the percentage of cells in the G1 stage (from 40.34% to 38.34%), significantly increased the percentage of cells in the S stage (from 31.36% to 36.28%), and significantly reduced the percentage of cells in the G2/M stage (from 28.30% to 25.38%) (Figure 3A,B).

To further determine the effect of DNA replication after *BmCDK5* overexpression, the cells were labelled with 5-bromodeoxyuridine (EdU). The number of EdU-labeling cells was significantly higher in cells overexpressing *BmCDK5* compared to the control (Figure 3C,D); this indicates that the relative rate of DNA synthesis was greatly increased after overexpressing *BmCDK5*. Meanwhile, the proliferation of cells was analyzed using a cell counting kit-8 (CCK-8) assay; the results showed that overexpression of *BmCDK5* promoted the proliferation of cells at 72 h post-transfection (Figure 3E). Therefore, all results indicate that overexpression of *BmCDK5* accelerates the mitotic period (M) process of the cell cycle and promotes cell proliferation.

### 3.4. Knock out BmCDK5 Gene Inhibiting Cell Proliferation in BmNS Cells

To further verify that *BmCDK5* is involved in cell cycle regulation, CRISPR/Cas9 technology was used to knock out this gene. We transfected the recombinant plasmid to knock out *BmCDK5* (BmCDK5-KO) in cells and assessed the knockout efficiency of *BmCDK5* at 48 h after transfection. The result showed that *BmCDK5* was successfully edited (Appendix A).

In addition, the number of EdU-labeling cells was significantly lower in cells with *BmCDK5* knocked out compared to the control (Figure 4A,B), which indicates that the relative rate of DNA synthesis after *BmCDK5* knockout is significantly reduced. The CCK-8 assay also showed that knockout of *BmCDK5* inhibited the proliferation of cells 48 h post-transfection (Figure 4C). Therefore, all results indicate that knocking out *BmCDK5* can inhibit cell proliferation.

### 3.5. BmCDK5 Interacts with BmCNN

The phylogenetic analysis of *CNN* showed that *BmCNN* clustered together with *Plutella ylostella CDK5* regulatory subunit associated protein (Appendix A). *BmCNN* has two spliceosomes; functions similar to those in our previous studies were found, with differences only in length compared to our previous studies [32]. Therefore, we cloned *BmCNN* with a longer sequence in this study, to explore its relationship with *BmCDK5*. In addition, we co-transfected HA-BmCNN and Flag-BmCDK5 overexpressed plasmids into BmNS cells. Immunofluorescence (IF) was used to analyze their localization, which showed that BmCNN was expressed only in the nucleus, BmCDK5 was expressed both in cytoplasm and the nucleus, and they were co-localized in the cytoplasm (Figure 5A). Following this, we analyzed the fluorescence signal of the yellow line profile. The *X*-axis represents the signal position and the *Y*-axis represents the signal strength. It was seen that BmCDK5 and BmCNN signal positions coexist simultaneously, indicating the existence of co-location (Figure 5B). Next, Co-Immunoprecipitation (Co-IP) experiments were performed using anti-HA and anti-Flag monoclonal antibodies, which showed interaction between BmCNN and BmCDK5 (Figure 5C). The IF and Co-IP results both show that BmCDK5 interacts with BmCNN.

To confirm whether there was a relationship between the expression pattern of *BmCDK5* and *BmCNN*, we analyzed the expression profiles of *BmCNN*. *BmCNN* was also highly expressed in the S phase (Figure 5D). The expression levels of *BmCNN* were examined in the hemolymph, epidermis, head, testis, ovary, malpighian tubule, trachea, midgut, fat body and silk gland of the third day of fifth instar larvae using qRT-PCR. The results showed that *BmCNN* was also highly expressed in the gonads (Figure 5E), as was *BmCDK5* (Figure 2A). Additionally, the expression pattern of *BmCNN* during the silkworm growth period was consistent with that of *BmCDK5*, which was highly expressed in the third day of fourth instar larvae, in the early and after wandering stages of the growth period (Figure 5F).

### 3.6. BmCDK5 and BmCNN Both Affect the Cytoskeleton Morphology of BmNS Cells

The effects of *BmCDK5* and *BmCNN* overexpression on cytoskeleton morphology were verified by transfecting HA-BmCNN and Flag-BmCDK5 recombinant plasmids into the cells. IF was conducted after 48 h of cell transfection; the results show an abnormal cytoskeleton morphology which was caused by the overexpression of *BmCDK5* and *BmCNN* (Figure 6A). Meanwhile, the expression levels of α-Tubulin did not change compared to the control groups (Figure 6B). The results suggest that overexpression of *BmCDK5* and *BmCNN* did not induce changes in microtubule protein expression, but significantly changed microtubule morphology.

In addition, to determine the effect of DNA replication after *BmCDK5*-*BmCNN* complex overexpression, the complex transfected cells were labelled with EdU (Figure 7A,B). The number of EdU-labelling cells was drastically higher in cells overexpressing *BmCDK5*-*BmCNN* complex compared to the control. Interestingly, the knockout groups had the opposite effect (Figure 7C,D), which indicates that expression of the *BmCDK5*-*BmCNN* complex greatly increased the relative rate of DNA synthesis.

## 4. Discussion

As a CDKs, CDK5 plays an important role in regulating the progress of cell cycles and cell division [11,33]. However, the regulation mode of CDK5 differs from that of other CDKs, which mainly control cell growth by affecting changes in the cytoskeleton [34]. There are few studies of CDK5 in insects compared to mammals and humans. Studies of CDK5 in *B. mori* are crucial due to this insect’s economic importance and because it is a model of the order Lepidoptera [35]. Several cyclins and CDK proteins are identified in *B. mori*, such as cyclin B, cyclin E, CDK1, CDK2, and CDK4 [36,37]; however, little is known about the function of CDK5 in *B. mori*. In this article, we systematically analyzed the function of the *BmCDK5* gene, and demonstrated that *BmCDK5* is involved in the cell cycle of the silkworm through interaction with BmCNN to regulate cytoskeleton structure.

CDK5 gradually became a popular research topic amongst scholars due to its role in the regulation of the cell cycle and its regulatory mechanism, which are different from those of other CDKs [7,8]. In the past, it was reported that CDK5 had no function in regulating the cell cycle [9,10]. Further studies of CDK5 and its regulatory proteins were carried out and the results revealed protein interactions between CDK5 and CyclinD1, D3, and Cyclin l. These combinations activated the protein kinase and were indirectly involved in cell cycle regulation [38,39]. In our study, overexpression of the BmCDK5 gene accelerates the process of the mitotic period (M) of the cell cycle and promotes cell proliferation. Knocking out the BmCDK5 gene inhibited cell proliferation. Our results proved that *BmCDK5* plays an important role in regulating the cell cycle and promotes cell proliferation in BmNS cells.

Further to this, we analyzed the expression of BmCDK5 in different cell cycle phases, and found that BmCDK5 was highly expressed in S phase. CDK5 expression and its influence on the cell cycle were analyzed comprehensively. It is inferred that CDK5 is highly expressed and accumulated in the S phase, which accelerates the process of the M phase and promotes the cell cycle. Analysis of CDK5 expression in various tissues showed that BmCDK5 was expressed in all larval tissues, with the highest expression level in the gonads. CDK5 was highly expressed in the third day of fourth instar larvae, in the early and after wandering stages of the growth period. As a cyclin-dependent kinase, BmCDK5 was highly expressed in silkworm testes and ovaries, which indicates that BmCDK5 is particularly necessary for cell proliferation of the reproductive organs.

Previous studies proved that CDK5 plays an important role in neurite outgrowth, neuronal migration, and neuronal cell survival [40,41]. CDK5 contributes to the migration of the developing central nervous system, the cytoskeleton structure, and organization of the nerve cell [13]. CDK5 plays an important role in cytoskeletal regulation and is closely associated with tubulin [13,19,20]. Moreover, microtubules are highly dynamic polymers that form a major filament system of the cytoskeleton in eukaryotic cells [22]. Centrosomin (CNN) is a core component in mitotic centrosomes during syncytial development, which is abundantly distributed in the PCM [25,26]. We identified a protein called BmCNN that can interact with BmCDK5, and follows the same trends as *BmCDK5* both in the cell cycle phase and in the space-time expression pattern of *B. mori*. *BmCDK5* and *BmCNN* can regulate cell proliferation, and affect cytoskeleton morphology, but they do not induce changes in the quantity of cytoskeleton protein expression.

## 5. Conclusions

In conclusion, we identified a homologous gene of the Cyclin-dependent kinase family, BmCDK5, in the silkworm, *B. mori*. We verified *BmCDK5* expression patterns at different developmental stages and in several tissues. Additionally, we determined the expression patterns of *BmCDK5* in different phases of the BmNS cell cycle. Moreover, we obtained evidence for the interaction between BmCDK5 and BmCNN. Based on our results, it is speculated that BmCDK5 interacts with BmCNN to regulate cell proliferation by affecting microtubule morphology in BmNS cells. Our findings provide insight into the relationship between the cell cycle functions of the Cyclin-dependent kinase family and cytoskeleton morphology.

## Figures and Tables

**Figure 1 insects-13-00609-f001:**
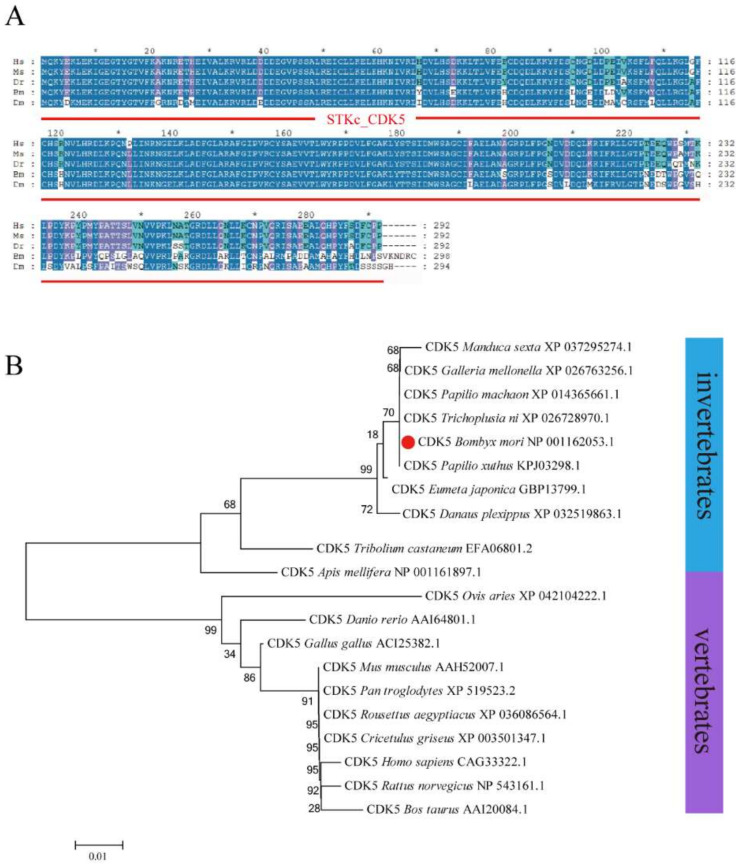
Homology and evolution analysis of BmCDK5. (**A**) Multiple sequence alignment of *CDK5* from *Homo sapiens*, *Mus musculus*, *Danio rerio*, *Bombyx mori* and *Drosophila melanogaster*, containing a conservative CDK5 domain (red line). (**B**) Phylogenetic tree of *CDK5* from invertebrates and vertebrates; BmCDK5 is marked by red origin. * 0.01 ≤ *p* < 0.05.

**Figure 2 insects-13-00609-f002:**
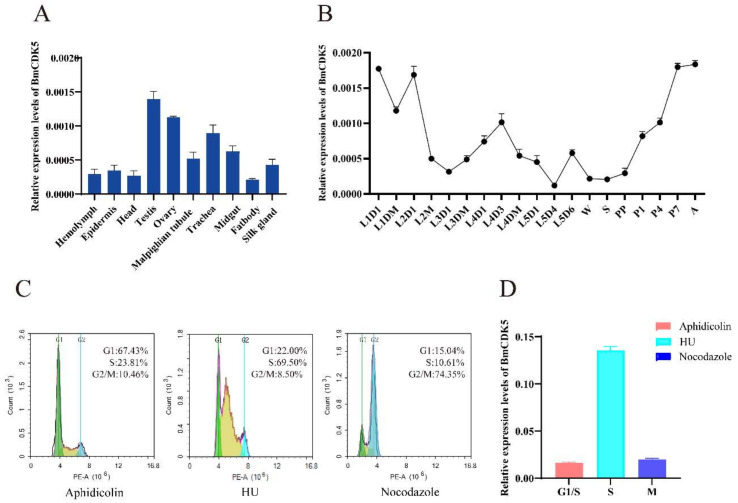
Expression patterns of *BmCDK5*. (**A**) The expression patterns of *BmCDK5* in different tissues of *B. mori* in fifth instar larvae. (**B**) The expression patterns of *BmCDK5* in different developmental stages of *B. mori*. (**C**) BmNS cells were blocked by synchronous reagents: Aphidicolin for the first gap phase of the cell cycle (G1); Hydroxyurea (HU) for the DNA synthesis phase of the cell cycle (S); and Nocodazole for the second gap phase and the mitotic phase of the cell cycle (G2/M). (**D**) The expression patterns of *BmCDK5* in different phases of the cell cycle in BmNS cells.

**Figure 3 insects-13-00609-f003:**
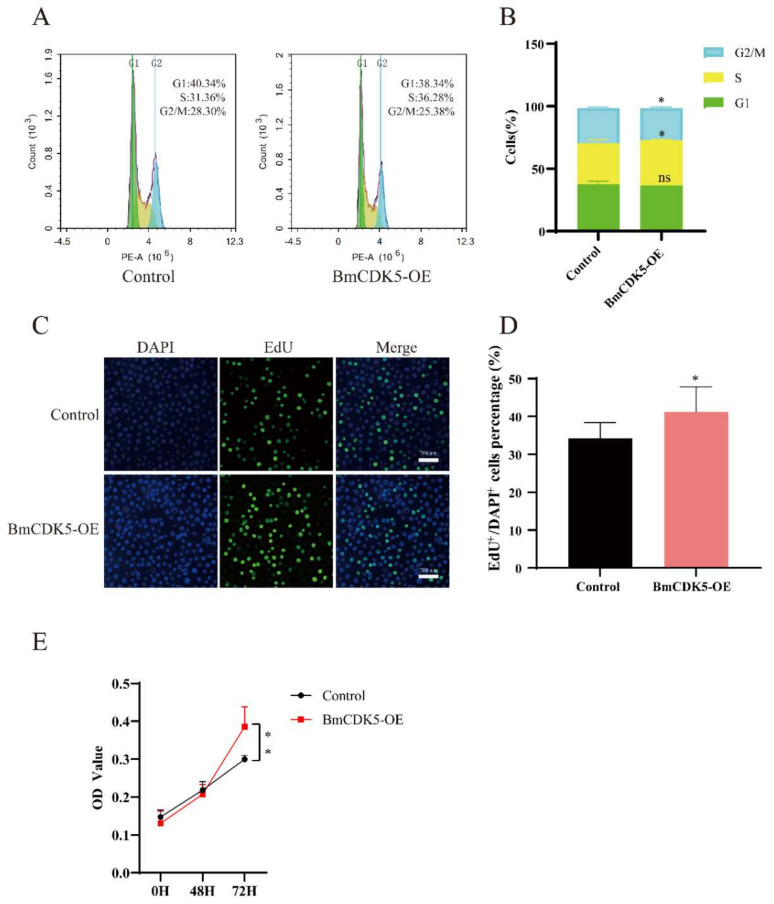
Overexpression of *BmCK5* gene in BmNS cells. (**A**) Cell cycle analysis of BmNS cells transfected with control plasmids or pIZ-*BmCDK5* plasmids using flow cytometry. (**B**) Statistical analysis of cell percentages at G1, S, and G2/M stages. (**C**) EdU labeling of BmNS cells. Green fluorescence represents EdU positive or EdU^+^ cells. Blue fluorescence represents nuclei stained with DAPI. (**D**) Statistical analysis of EdU positive or EdU^+^ cells. (**E**) The CCK-8 assay was performed to detect cell proliferation ability at 0 h, 48 h, and 72 h (ns *p* ≥ 0.05, * 0.01 ≤ *p* < 0.05, ** *p* < 0.01).

**Figure 4 insects-13-00609-f004:**
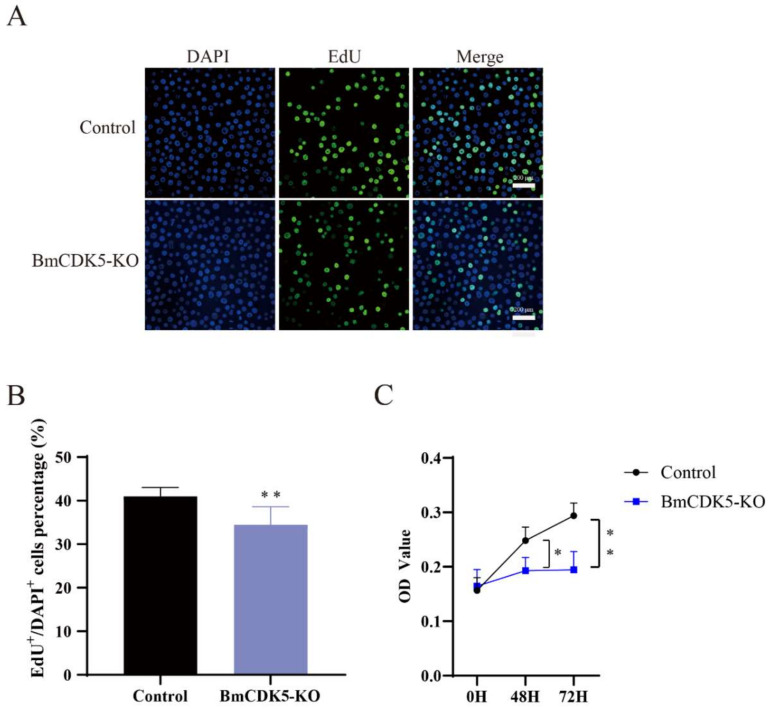
Knocked out *BmCK5* gene in BmNS cells. (**A**) EdU labeling BmNS cells. Green fluorescence represents EdU positive or EdU^+^ cells. Blue fluorescence represents nuclei stained with DAPI. (**B**) Statistical analysis of EdU positive or EdU^+^ cells. (**C**) The CCK-8 assay was performed to detect cell proliferation ability at 0 h, 48 h, and 72 h (* 0.01 ≤ *p* < 0.05, ** *p* < 0.01).

**Figure 5 insects-13-00609-f005:**
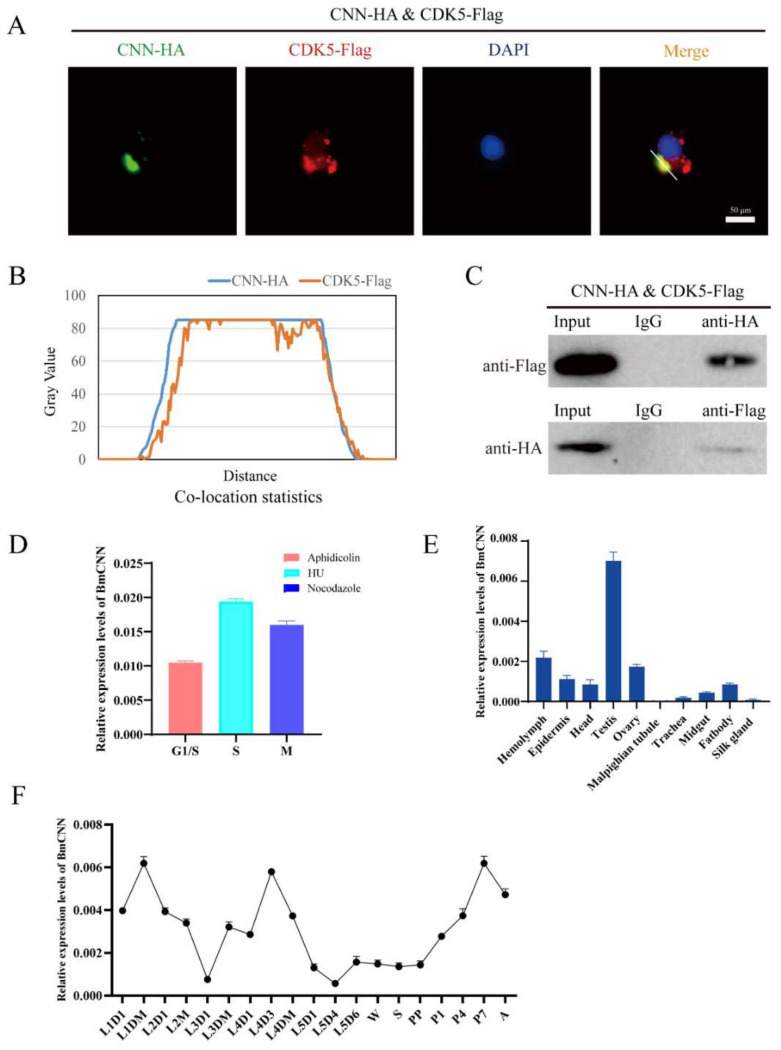
Interaction of BmCDK5 and BmCNN. (**A**) The co-localization of BmCNN (green) with BmCDK5 (red) using IF analysis. (**B**) The statistics analysis of IF co-localization. (**C**) The interaction of BmCDK5 and BmCNN by Co-IP assays. (**D**) The expression pattern of *BmCNN* in different phase of the cell cycle in BmNS cells. (**E**) The expression pattern of *BmCNN* in different tissues of *B. mori* in fifth instar larvae. (**F**) The expression pattern of *BmCNN* in different development stages of *B. mori*.

**Figure 6 insects-13-00609-f006:**
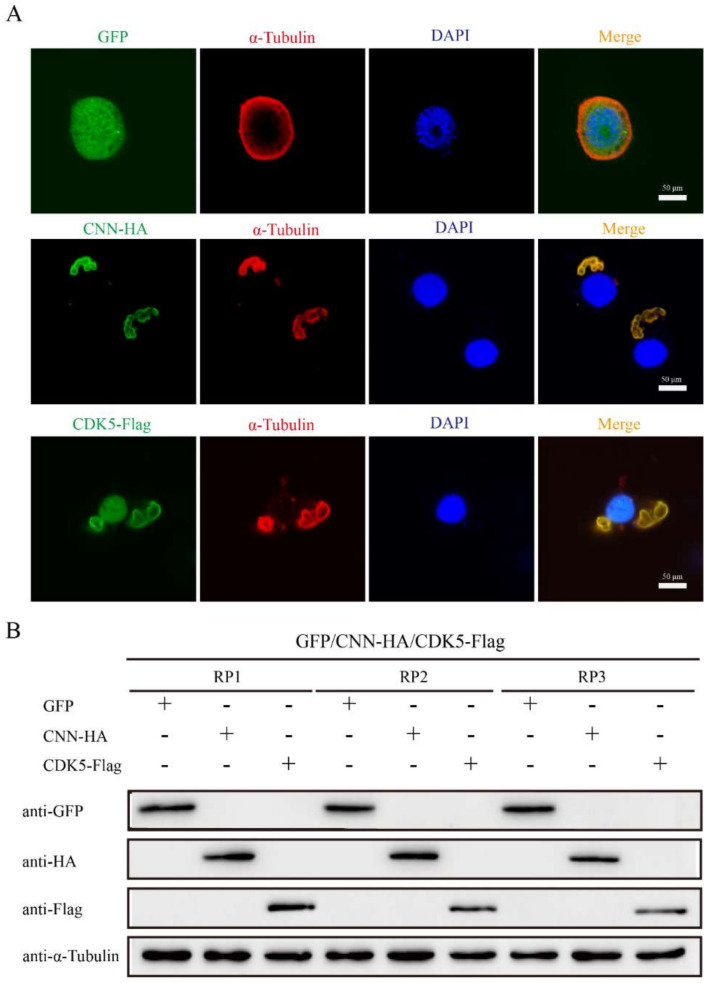
Effect of BmCDK5-BmCNN complex on cytoskeletal morphology. (**A**) IF analysis of cytoskeletal changes after overexpressing BmCNN and BmCDK5. Cytoskeleton was labeled using α-Tubulin (red). (**B**). Western blotting analysis of cytoskeletal changes after overexpressing BmCNN and BmCDK5. Three biological repeats, including RP1, RP2, and RP3.

**Figure 7 insects-13-00609-f007:**
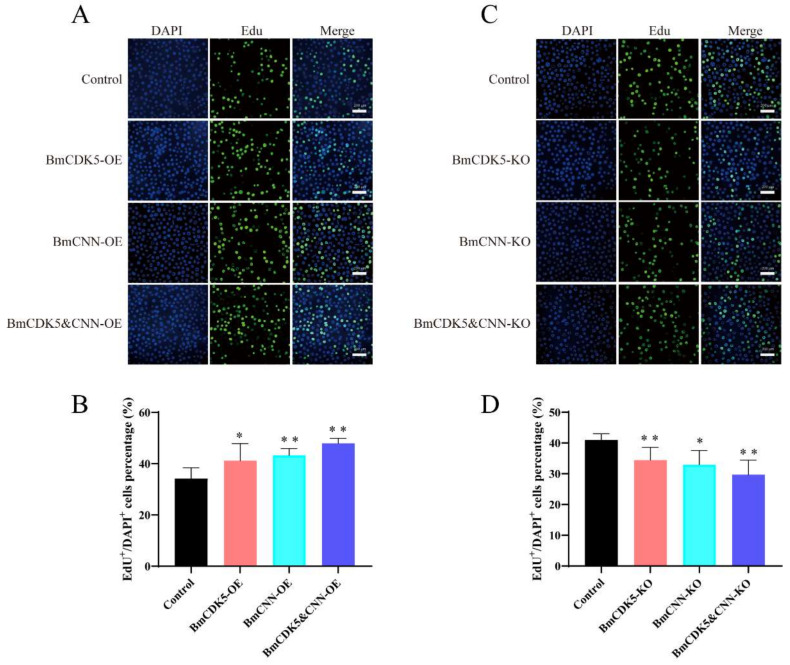
Cell proliferation regulated by BmCDK5-BmCNN complex in BmNS cells. (**A**,**C**) The EdU labeling experiment in BmNS cells. (**B**,**D**) Statistical analysis of EdU positive or EdU^+^ cells (* 0.01 ≤ *p* < 0.05, ** *p* < 0.01).

## Data Availability

Data is contained within the article and Appendix A.

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
