# Peer review of "BmCDK5 Affects Cell Proliferation and Cytoskeleton Morphology by Interacting with BmCNN in Bombyx mori"

_insects, 2022, doi:10.3390/insects13070609_

Round 1

Reviewer 1 Report

Comments:

insects-1776588 BmCDK5 affects cell proliferation and cytoskeleton morphology by interacting with BmCNN in Bombyx mori

In this manuscript, Wei and Zhou et al. identified BmCDK5, a silkworm homologous gene of the Cyclin-dependent kinase family and studied its functions in cell proliferation and cytoskeleton morphology. They also identified its interacted protein BmCNN in promoting cell proliferation and regulating the cytoskeleton morphology. The manuscript is written concisely and will be interesting for scientists in the insect field. I recommend acceptation of the manuscript with minor revision. 

Minor comments:

Line 14: replace ‘expressed highly’ with ‘highly expressed’ 

Line 16-17: replace ‘Knocked out of BmCDK5 gene led to the cell proliferation had been inhibited.’ with ‘Knocked out of BmCDK5 gene inhibited cell proliferation.’

Line 17: replace ‘we identified a protein called BmCNN which can…’ with ‘we identified a protein, BmCNN, which can…’

Line 18: replace ‘same express trends’ with ‘same express patterns’

Line 19: replace ‘space-time expression pattern’ with ‘spatial-temporal expression’

Line 21: replace ‘induce changes in microtubule protein expression quantity’ with ‘induce expression changes of microtubule protein’

Line 49: replace ‘Current research’ with ‘Current study’

Line 57: replace ‘which important for’ with ‘which is important for’

Line 65-66: replace ‘a homologous gene of the Cyclin-dependent kinase family, BmCDK5, has been identified in B.mori.’ with ‘we identified a silkworm homologous gene of the Cyclin-dependent kinase family, BmCDK5.’

Line 73: replace ‘national center for biotechnology information’ with ‘National Center for Biotechnology Information’

Line 74: replace ‘alignment was’ with ‘alignments were’

Line 75: replace ‘domain was’ with ‘domains were’

Line 77: it should have a website link or reference for ‘CRISPR direct’.

Line 81: The authors should give the more detail information about ‘same kind of restriction enzymes’, which enzymes used here.

Line 84-88: The authors should provide more detail method for the cell transfection because this manuscript did most of experiments in cell line. And the experiments repeatability is requested for other readers. For example, the concentration of cells and vectors they used, the transfection time, etc.

Line 110: replace ‘was incubated’ with ‘was incubated with’

Line 148: The authors mentioned that BmCDK5 is highly conserved, they should provide the ‘conserved’ information, in animal? in insect? or all species?

Figure 1. in (A) there are some Arabic numerals in the protein sequences, such as ‘MQKYeK6E’, can the authors explain what’re they mean? In (B), species names should italic.

Line 152: replace ‘Homology analysis of BmCDK5.’ with ‘Homology and evolution analysis of BmCDK5.’

Line 160: replace ‘in the gonads’ with ‘in gonads’

Line 200, 217: replace ‘Nuclei’ with ‘nuclei’

Line 204: replace ‘CRISPER/Cas9’ with ‘CRISPR/Cas9’

Line 208: replace ‘(Fig. S B)’ with ‘(Fig. S1C)’

Line 211: replace ‘greatly reduced’ with ‘significantly reduced’ 

Line 213: replace ‘all the results’ with ‘all results’

Line 222: ‘(Fig. S C)’ should ‘(Fig. S1A)’?

Line 300: replace ‘expressed highly’ with ‘highly expressed’ 

Line 304: replace ‘in the gonads’ with ‘in gonads’

Supplemental Material: in (A) species names should italic. (B) it should be better if the authors split to two images. 

Author Response

Dear respected reviewer

We are very grateful for the comments and suggestions from you. Thank you very much for your consideration and time in reviewing our manuscript. We have made thorough check and have made corrections based on suggestions given. Here we have listed all the answers for the questions given.

Point 1: Line 14: replace ‘expressed highly’ with ‘highly expressed’ 

Response 1: Thank you very much for your suggestion, and we have corrected it in this manuscript including line 14.

Point 2: Line 16-17: replace ‘Knocked out of BmCDK5 gene led to the cell proliferation had been inhibited.’ with ‘Knocked out of BmCDK5 gene inhibited cell proliferation.

Response 2: Thank you very much for your suggestion, and we have corrected it on line 16 and 361.

Point 3: Line 17: replace ‘we identified a protein called BmCNN which can…’ with ‘we identified a protein, BmCNN, which can…’

Response 3: Thank you very much for your suggestion, and we have corrected it on line 17.

Point 4: Line 18: replace ‘same express trends’ with ‘same express patterns’

Response 4: Thank you very much for your suggestion, and we have corrected it on line 18.

Point 5: Line 19: replace ‘space-time expression pattern’ with ‘spatial-temporal expression’

Response 5: Thank you very much for your suggestion, and we have corrected it on line 18.

Point 6: Line 21: replace ‘induce changes in microtubule protein expression quantity’ with ‘induce expression changes of microtubule protein’

Response 6: Thank you very much for your suggestion, and we have corrected it on line 20、78.

Point 7: Line 49: replace ‘Current research’ with ‘Current study’

Response 7: Thank you very much for your suggestion, and we have corrected it on line 58.

Point 8: Line 57: replace ‘which important for’ with ‘which is important for’

Response 8: Thank you very much for your suggestion, and we have corrected it on line 66.

Point 9: Line 65-66: replace ‘a homologous gene of the Cyclin-dependent kinase family, BmCDK5, has been identified in B.mori.’ with ‘we identified a silkworm homologous gene of the Cyclin-dependent kinase family, BmCDK5.’

Response 9: Thank you very much for your suggestion, and we have corrected it on line 74.

Point 10: Line 73: replace ‘national center for biotechnology information’ with ‘National Center for Biotechnology Information’

Response 10: Thank you very much for your suggestion, and we have corrected it on line 82.

Point 11: Line 74: replace ‘alignment was’ with ‘alignments were’

Response 11: Thank you very much for your suggestion, and we have corrected it on line 83.

Point 12: Line 75: replace ‘domain was’ with ‘domains were’

Response 12: Thank you very much for your suggestion, and we have corrected it on line 84.

Point 13: Line 77: it should have a website link or reference for ‘CRISPR direct’.

Response 13: Thank you very much for your suggestion, and we have corrected it on line 86.

Point 14: Line 81: The authors should give the more detail information about ‘same kind of restriction enzymes’, which enzymes used here.

Response 14: Thank you very much for your suggestion, and we have corrected it on line 90.

Point 15: Line 84-88: The authors should provide more detail method for the cell transfection because this manuscript did most of experiments in cell line. And the experiments repeatability is requested for other readers. For example, the concentration of cells and vectors they used, the transfection time, etc.

Response 15: Thank you very much for your suggestion, and we have corrected it on line 104、129、136、139 and 158.

Point 16: Line 110: replace ‘was incubated’ with ‘was incubated with’

Response 16: Thank you very much for your suggestion, and we have corrected it on line 148.

Point 17: Line 148: The authors mentioned that BmCDK5 is highly conserved, they should provide the ‘conserved’ information, in animal? in insect? or all species?

Response 17: We are sorry about the unclear description, and we have corrected it in “BmCDK5 is highly conserved in animal” on line 195.

Point 18: Figure 1. in (A) there are some Arabic numerals in the protein sequences, such as ‘MQKYeK6E’, can the authors explain what’re they mean? In (B), species names should italic.

Response 18: We are very sorry about the mistakes, and we have corrected it in Fig 1.

Point 19: Line 152: replace ‘Homology analysis of BmCDK5.’ with ‘Homology and evolution analysis of BmCDK5.’

Response 19: Thank you very much for your suggestion, and we have corrected it on line 200.

Point 20: Line 160: replace ‘in the gonads’ with ‘in gonads’

Response 20: Thank you very much for your suggestion, and we have corrected it on line 211.

Point 21: Line 200, 217: replace ‘Nuclei’ with ‘nuclei’

Response 21: Thank you very much for your suggestion, and we have corrected it on line 165、255 and 275.

Point 22: Line 204: replace ‘CRISPER/Cas9’ with ‘CRISPR/Cas9’

Response 22: Thank you very much for your suggestion, and we have corrected it on line 265.

Point 23: Line 208: replace ‘(Fig. S B)’ with ‘(Fig. S1C)’

Response 23: Thank you very much for your suggestion, and we have corrected it on line 269.

Point 24: Line 211: replace ‘greatly reduced’ with ‘significantly reduced’

Response 24: Thank you very much for your suggestion, and we have corrected it on line 272.

Point 25: Line 213: replace ‘all the results’ with ‘all results’

Response 25: Thank you very much for your suggestion, and we have corrected it on line 250 and 274.

Point 26: Line 222: ‘(Fig. S C)’ should ‘(Fig. S1A)’?

Response 26: Thank you very much for your suggestion, and we have corrected it on line 288.

Point 27: Line 300: replace ‘expressed highly’ with ‘highly expressed’

Response 27: Thank you very much for your suggestion, and we have corrected it on line 305 and 310.

Point 28: Line 304: replace ‘in the gonads’ with ‘in gonads’

Response 28: Thank you very much for your suggestion, and we have corrected it on line 211、308 and 377.

Point 29: Supplemental Material: in (A) species names should italic. (B) it should be better if the authors split to two images.

Response 29: We are very sorry about the mistakes, and we have corrected it in Fig S1.

Reviewer 2 Report

The manuscript by Wei et al identified a homologous gene of the Cyclin-dependent kinase family, BmCDK5, in Bombyx mori, performed overexpression and knockout experiments of this gene, and identified the BmCNN which can interact with BmCDK5. Given that the importance of silkworms as an agricultural insect and a research model for Lepidopteran, and importance of cell cycle study in the proliferation and differentiation of living organisms, this work is interesting and will provide some useful information to this field. Therefore, I feel that this manuscript is suitable for this journal if the authors could make the following improvements in a possible revision.

1. Figure 2 and Figure 3, the images of cell cycle analysis are not clear enough, the authors should provide high resolution images.

2. Figure 4, the CRISPR editing results are important evidences to support the conclusion, the authors are suggested to move these results from the supplementary materials to the main figure.

3. Figure 2 and Figure 5, silk-gland should be silk gland.

4. line 29, in generally should be in general.

5. line 57, which important should be which is important.

6. line 61, abundantly distributes should be is abundantly distributed.

7. lines 76-77, The sgRNAs for knock out cloning vector were designed by CRISPR direct should be the sgRNAs for knock-out cloning vectors were designed by CRISPR direct (website ?).

8. line 146, the source of Dazao strain should be indicated here or in the materials and methods section. 

Author Response

Dear respected reviewer

We are very grateful for the comments and suggestions from you. Thank you very much for your consideration and time in reviewing our manuscript. We have made thorough check and have made corrections based on suggestions given. Here we have listed all the answers for the questions given.

Point 1: Figure 2 and Figure 3, the images of cell cycle analysis are not clear enough, the authors should provide high resolution images.

Response 1: Thank you very much for your suggestion, and we have corrected it in Figure 2 and Figure 3.

Point 2: Figure 4, the CRISPR editing results are important evidences to support the conclusion, the authors are suggested to move these results from the supplementary materials to the main figure.

Response 2: Thank you very much for your suggestion. I I can't agree more with your suggestion and have considered it when writing. However, due to the sequence of exposition and picture arrangement of the whole article, I couldn't put it in the attachment at last.

Point 3: Figure 2 and Figure 5, silk-gland should be silk gland.

Response 3: Thank you very much for your suggestion, and we have corrected it in Figure 2 and Figure 5.

Point 4: line 29, in generally should be in general.

Response 4: Thank you very much for your suggestion, and we have corrected it on line 28.

Point 5: line 57, which important should be which is important.

Response 5: Thank you very much for your suggestion, and we have corrected it on line 66.

Point 6: line 61, abundantly distributes should be is abundantly distributed.

Response 6: Thank you very much for your suggestion, and we have corrected it on line 70.

Point 7: lines 76-77, The sgRNAs for knock out cloning vector were designed by CRISPR direct should be the sgRNAs for knock-out cloning vectors were designed by CRISPR direct (website ?).

Response 7: We are sorry about the unclear description, and we have corrected it on line 86.

Point 8: line 146, the source of Dazao strain should be indicated here or in the materials and methods section.

Response 8: We are sorry about the unclear description, and we have corrected it on line 95-98.

This manuscript is a resubmission of an earlier submission. The following is a list of the peer review reports and author responses from that submission.